# Quality of Chicken Fat by-Products: Lipid Profile and Colour Properties

**DOI:** 10.3390/foods9081046

**Published:** 2020-08-03

**Authors:** Lina María Peña-Saldarriaga, Juana Fernández-López, José Angel Pérez-Alvarez

**Affiliations:** 1Research & Development Department, Bios Group, Cra 48 No. 274 Sur-89 Envigado, 055422 Antioquia, Colombia; lina.pena@grupobios.co; 2IPOA Research Group, Agro-Food Technology Department, Higher Polytechnic School of Orihuela, Miguel Hernández University, Orihuela, 03312 Alicante, Spain; j.fernandez@umh.es

**Keywords:** chicken fat by-products, unsaturated fatty acids, colour properties, lipid profile

## Abstract

The growth in the production and consumption of chicken meat and related products is responsible for the formation a large number of by-products. Among these, abdominal and gizzard fat is usually considered as waste and thus is discarded, creating an environmental problem. This work aims to characterize chicken fat by-products, evaluating their lipid profile and colour properties for their potential use as fat sources in meat products in substitution of traditionally used fats. In addition, the role of farm location, keeping the feeding and other farmer routines fixed, in the lipid profile was also evaluated. “Parrilleros” Colombian chickens from three farms located in various geographical zones of the Antioquia region were selected. After slaughtering, abdominal and gizzard fat was obtained. Lipid profile was evaluated by gas chromatography and the CIELAB colour properties were assessed. The production results and the lipid profile of chicken fat by-products (abdominal and gizzard fat) was similar in the three farms studied, which is important for their potential application as fat source in the formulation of meat products. The predominant fatty acids were oleic, palmitic and linoleic acids, showing a higher amount of unsaturated fatty acids than the fat sources traditionally used for this purpose. Valorization of chicken by-products by the use of abdominal and gizzard fat as fat source in chicken meat products formulation could be a feasible alternative contributing to the poultry sector sustainability.

## 1. Introduction

Poultry farming has been the main impetus for the sustained and steady economic development of Colombian agriculture in recent years, and is considered a determining variable in the growth of the Gross Domestic Product of the agricultural sector in the country. The growth in the poultry industry in Colombia is mainly due to increased domestic consumption. A decade ago the per capita consumption of poultry meat in Colombia was about 23 kg of chicken meat per year, while today it is 35.5 kg [1]. The industry has developed to such an extent that poultry products are now the most important source of animal protein in Colombia (contributing 50%), a trend that underlines the importance of this industry in the country and its constant growth. Not only in Colombia is the poultry industry important, but it also plays a relevant role in feeding much of the rest of the world. According to the Organization for Economic Cooperation and Development (OECD) and the Food and Agricultural Organization (FAO), the worldwide per capita consumption of chicken meat in the last decade has increased by 15%, growth which has outstripped that registered for beef and pork. The main consumers are the United States and Brazil, whose annual consumption exceeds 40 kg per capita [2]. Such an increase in the consumption of chicken meat is mainly due to the perception by health-conscious consumers that chicken meat is a low-fat source of healthy nutrition, rich in unsaturated fat and a high in protein [3]. In addition, chicken meat is increasingly used in the development of new chicken-based convenience products (chicken bologna, chicken nuggets, chicken hotdogs, chicken wings), which have been successfully marketed for consumption at home and also in the growing fast-food industry [4].

However, the rapid growth of poultry production has led to the massive generation of food-processing by-products likes bones, viscera, abdominal fat, feet, head, blood and feathers. If these by-products were regarded as having greater nutritional value, their use would contribute to the development of a sustainable food industry while increasing the value of this sector [5]. Until now, these by-products have only been sold as animal feed and to pet food processors [6,7,8] and, recently, for the production of biodiesel [9]. However, there are no references about the possible use of some of these by-products as raw materials for use in human food processing. For example, it may be possible that the abdominal and gizzard fat that remains inside the poultry carcass, where it represents approximately 2–2.5% of the total weight of the slaughtered chicken [10], could be used as fat source for the production of chicken sausages or other meat products, especially taking into account its characteristic content of unsaturated fatty acids. Until now, this abdominal and gizzard fat has been discarded by small producers, together with the viscera, feathers and blood, thus creating and environmental problem.

The production of high quantities of by-products by the poultry industry and the potential of abdominal and gizzard fat as a healthy fat source in different applications, about which little information is available, led to the development of this study. The main objective was to determinate the fatty acid profile and colour properties of poultry fat by-products (in this case, abdominal and gizzard fat) and to assess whether these properties remain stable and whether they depend on the farm conditions (feeding and geographical location).

## 2. Materials and Methods 

### 2.1. Experimental Design: Animals and Diets

One-day-old “Parrilleros” Colombian chickens from three commercial farms (La Nirvana, La Goleta and Villa Rita) located in various geographical zones of the Antioquia region in Colombia (Barbosa, Yolombó and Caldas, respectively), characterized by their different climatic conditions, were reared on litter floors (wood shavings) in open-sided housing conditions with feed and water provided *ad libitum*. The average number of birds reared and the average density (bird/m^2^) in each farm were: Villa Rita, 78,432 and 12.9, respectively; La Nirvana, 314,200 and 13.0, respectively; and La Goleta 1,132,002 and 12.4, respectively. In all the farms the photoperiod was 12 h (±30 min) (12L/12D). In each farm, 75 birds were selected for the experiment (kept in pens on litter) and were divided into 3 replications with 25 birds per group. Each bird had a padlock badge for identification during measurements. All the chickens were initially fed the same balanced diet: a “pre-starter” diet until they reached 150 g in weight and a “starter” diet until 900 g (approx. 16 days); this was followed by a “finisher” diet based on standard formulations used in different fattening periods, until slaughter at 45 days of age. Nine different “finisher” diets were assessed (Table 1) depending on the availability of raw materials and prices in attempt to minimize costs for the companies, while maintaining the same nutritional levels. The finisher experimental diets and water were offered *ad libitum*. 

Productivity parameters (final body weight (FBW), daily body weight gain (BWG), feed intake (FI), and feed conversion ratio (FCR)) were monitored and recorded for the entire flock (75 birds) per farm. At 45 days, thirteen chickens per farm, with an FBW close to the mean of the whole group were slaughtered in an abattoir (previous electrical stunning) licensed by The National Institute for the Surveillance of Drugs and Foods (Colombia) and abdominal and gizzard fat was obtained (Figure 1). The fat samples were refrigerated and sent to the Food Science and Technology Institute laboratory to assess their fatty acid content and colour.

### 2.2. Chemical analysis

Samples of fat (200 g; 65% abdominal fat and 35% gizzard fat, the normal fat proportions of the carcass) were dried and extracted following the Soxhlet procedure and using diethyl ether as the extraction solvent [11]. The methyl esters from fatty acids (FAME) were prepared using BF_3_ in methanol and stored at −80 °C until chromatographic analysis.

The FAME were analysed using a gas chromatograph (GC-2014 Gas Chromatograph, Shimadzu, Chiyoda-ku, Tokyo, Japan) equipped with a flame ionization detector, a split/splitless injector, and a fused silica capillary column containing polyethylene glycol as stationary phase (db-wax, 60 m × 0.25 mm, J&W Scientific, Santa Clara, CA, USA). The injector temperature was set to 230 °C. The initial column temperature was 80 °C for 2 min at a rate of 3 °C per minute, was raised to 180 °C at 30 °C per minute and was kept at this temperature for 30 min. After this time, the temperature was increased to 200 °C at a rate of 3 °C per minute and remained at this temperature for 108 min. The fatty acids were quantified using C11:0 methyl ester as internal standard. Identification of fatty acids was performed by comparison of the retention times with those of known fatty acids and the results expressed as percentage of the area of each fatty acid over the total area of fatty acids (%).

### 2.3. Colour Properties

The CIELAB space was chosen for colour determination following American Meat Science recommendations [12]. The following colour coordinates were determined: lightness (L*), redness (a*, ±red-green), and yellowness (b*, ± yellow-blue). The chroma saturation index [C* = (a*^2^ + b*^2^)^1/2^] and the hue angle (h* = tan^−1^ b*/a*) were also estimated. The reflectance spectra between 400 and 700 nm were also obtained at every 20 nm. These colour coordinates were determined by a SP62 spectrophotometer X-RITE (X-RITE, Grand Rapids, MI, USA). Measurements were made using D65 illuminant, 64 mm area and a 10° observer angle. These colour measurements were made in 43 samples of chicken fat by-products in their original solid form and also after heating at 78 °C for 3 min and re-solidifying at room temperature (re-solidified fat), simulating the thermal treatment applied for processing cooked meat products. 

### 2.4. Statistical Analysis 

The calculation of production results considered the entire flock, i.e., 75 birds in each farm. To determine the sampling of chicken fat, considering the chicken live weight and the farm of origin with different geographical location, confidence interval for one proportion—confidence interval Ross Lenth’s Piface- was used. The number of samples to be analysed for the lipid profile according to the statistical analysis performed on the sampling of fat in the plant was thirteen, for a variance of 6.6 obtained from the sum of the two types of chicken fat by-products. Colour data are reported as average ± standard deviation. The data were analysed statistically using IBM SPSS Statistics for Windows, version 23 (IBM Corp., Armonk, NY, USA).

## 3. Results

### 3.1. Chickens’ Performance

The production results (final body weight, average daily weight gain, feed intake and feed conversion ratio) were similar in the three farms (Table 2) and there were no statistically significant differences that depended on the feed used (*p* > 0.05).

### 3.2. Chemical Analysis

Importantly, there were no significant differences (*p* > 0.05) in the total fat content between the 8 different finisher diets used for chicken feed (Table 1). These diets were elaborated considering the composition and content of the ingredients used in each formula. In all the diets, linoleic (52.22%), oleic (24.87%) and palmitic (11.45%) fatty acids were identified as the predominant fatty acids. 

The weight of fat by-products per chicken carcass was approximately 40 g, of which 65% corresponded to abdominal fat and 35% to gizzard fat. The total yield for lipid extraction obtained in chicken fat by-products was 75%. 

Fatty acid profiles (% of total lipids) of chicken fat by-products from the 3 farms used in this study are shown in Table 3. No differences were found (*p* > 0.05) between the lipid profiles of chicken fat by-products from the 3 farms under study.

The predominant fatty acids in chicken fat by-products were oleic (C18:1), palmitic (C16:0) and linoleic (C18:2) acids (Table 3), which reflects the lipid profile of the diets (Table 1). The chicken fat by-products showed a higher unsaturated fat content (65.5%) than of saturated fat (30.3%), which also reflects the values of the diets (Table 1). 

### 3.3. Colour Properties

The colour parameters of chicken fat by-products are shown in Table 4. Solid fat had statiscally higher (*p* < 0.05) L* and a* values than the melted and re-solidified fat. By contrast, the b* coordinate, saturation index and hue values were higher (*p* < 0.05) when chicken fat by-products were previously melted. 

Figure 2 presents the reflectance spectra (400–700 nm) obtained for the solid fat and the melted and re-solidified fat. As it can be seen, the shape of the spectra for both types of fat is completely different. At all the wavelengths studied, solid fat showed higher (*p* < 0.05) reflectance percentages than the other fat. Melted and re-solidified fat did not show any reflectance from 400 to 480 nm (mainly corresponding to violet and blue), while from 480 to 540 nm (green) the reflectance values showed the higher increase (approx.13%), these reflectance values remaining constants until the end of the spectrum (corresponding to yellow, orange and red). 

## 4. Discussion

### 4.1. Chickens’ Performance

Our study found no significant effect of farm or finisher diet on the productivity parameters of chickens, and all the values obtained agree with the normal productive parameters reported for chickens reared in Antioquia (Colombia) [13].

### 4.2. Chemical Analysis

As explained above, the fat deposits of a chicken carcass come mainly from the diet, so that the lipid profile in these tissues reflect the lipid profile of the diet [14]. The interactions that take place between the nutrients that compose the diet and the synthesis and activity of lipogenic enzymes are responsible for a wide range of possibilities regarding lipid deposition in adipose tissue. Moreover, the biological activity of some fatty acids stimulates or inhibits specific lipogenic genes encoding enzymes [15].

The yields obtained for the lipid extraction in chicken fat by-products are markedly higher than the levels reported for chicken skin (<30%) [16,17], the usual chicken fat source in the meat industry. 

The fact that the lipid profile of chicken fat by-products from the 3 farms under study did not show differences is probably due to the stability of the feed used in each farm. Since the feeding and other farms’ routines were the same in all three farms, so that the only difference was their respective geographical location and climatic conditions, it seem safe to conclude that neither factor was important enough to modify this composition. This is very important because if chicken fat by-products are to be used as fatty ingredients in the meat industry, the greater the homogeneity in their composition, the easier it will be to formulate meat products. 

The lipid profile of fat by-products was within the range reported in the literature for chicken skin fat (Table 3) [17,18,19] which was to be expected because the lipid profile of different chicken carcass parts (skin, adipose tissue and meat) does not any show statistical differences [20]. 

Of other sources of animal fat commonly used in meat products (Table 5), chicken fat by-products have the highest amount of unsaturated fatty acids (UFA, 65.5%) and bovine tallow the lowest (44–50%). It must be noted that the chicken fat by-products analysed in this study contained a higher proportion of polyunsaturated fatty acids (PUFA, approx. 40% of total UFA) than pork or beef fat (has less than 20%). Unsaturated fatty acids include essential fatty acids that play beneficial roles in human health. Oleic acid may help decrease the circulating concentration of low density lipoprotein (LDL) cholesterol in humans and is considered a “healthy” fat [21]. High oleic acid values are desirable for their hypocholesterolemic action, and have the added advantage of not lowering high density lipoprotein (HDL) cholesterol (“good cholesterol”), and protecting against coronary heart diseases [22,23]. The essential fatty acids include the w3 and w6 families, which are not biologically synthetized by humans, but which are necessary for biological processes and therefore should be included in the human diet [23]. 

By contrast, the highest saturated fatty acid (SFA) levels are found in beef tallow (46–55%) and the lowest in poultry fat by-products (30.2%). Taking into consideration that the high consumption of saturated fatty acids has been associated with increased levels of serum cholesterol and LDL, both risk factors for cardiovascular diseases [26,27], using chicken fat by-products as fatty raw material in the meat industry could be considered advantageous. However, some studies suggest that the role of saturated fat in heart diseases is complex because of the heterogeneous biological effects of different saturated fatty acids and the diversity of food sources [27,28], so that not all SFAs should be considered hypercholesterolemic. These findings suggest that the specific matrix of different foods, including other fatty acids, nutrients, and bioactives, may biologically modify the effect of saturated fat in cardiovascular diseases. 

According to French et al. [29] the most undesirable fatty acid is myristic acid, which only represents 1.3% in chicken fat by-products (Table 3), 3% in beef tallow and 3.5% in pork backfat (Table 5). Several authors have reported that palmitic acid has a low hypercholesterolemic effect and stearic acid has no effect because it becomes oleic acid in the body [30] and so does not influence blood cholesterol levels. 

These results suggest that chicken fat can be used as fatty ingredient in formulating sausages, for example, as a partial or total substitute of traditional solid fat sources with their higher SFA concentrations, or be used together with chicken skin, thus increasing the amount of useful fat that can be obtained from poultry [31]. In addition, the high levels of UFA in chicken fat by-products could allow them to be used as frying oil as well as mixed with other solid fats to increase their plasticity. 

### 4.3. Colour Properties

The colour of foods is the first characteristic that makes an impression on consumers and is one of the most intuitive factors influencing consumer purchase decisions [32,33]. Contrary to what might be expected, pure fats and oils are colourless. The characteristic colours usually associated with some of them are imparted by foreign substances that are lipid-soluble and have been absorbed by these lipids. In the case of the fat from carcasses, the colour basically depends on the feed that the live animal received [34]. In the case of chickens, when maize (rich in carotenes and xanthophylls) is included in their diet, the fatty deposits take on a yellow colour. Another factor influencing fat colour is the concentration of haemoglobin retained in the capillaries of the adipose tissue and also the connective tissue that is included [35]. According to this author, mature adipose cells or adipocytes can easily reach a diameter of micron size and are almost filled by a single large droplet of triglyceride. Thus, the nucleus and cytoplasm of an adipose cell are restricted to a thin layer under the plasma membrane, which accounts for the low water content of fat. Mature adipose cells with very little cytoplasm contain few organelles. The large triglyceride droplet that fills most of the cell is not directly bounded by a membrane, but is restrained by a cytoskeletal meshwork of 10-nm filaments, which is most conspicuous in the adipose cells of poultry.

From a technological point of view, fat fulfils several functions in meat product processing (e.g., appearance, taste and textural properties) although, in the case of colour, its principal role is in the brightness of the resulting meat products. The colour coordinate values (L*, a* and b*) of the analysed chicken fat by-products are into the range reported by Sirri et al. [36] for chicken skin. These authors measured the colour coordinates in the skin of different parts of the chicken carcass (breast, thigh and shank) and reported the following values: 65.8-81.7 for lightness, −3.75–7.52 for redness, and 7.45–39.12 for yellowness. These data point to high variability in skin colour, especially in the case of b*, even taking into account that the total xanthophyll content of the feeds used was homogeneous (from 12 to 15 mg/kg of feed) in the different flocks studied. This suggests that, in addition to pigment concentrations, other factors could play an important role in determining the final skin colour of poultry.

The observed reduction in lightness and the increase in yellowness due to melting (Table 4) could be due to the reduction in moisture and the consequent increase in the concentration of yellow pigment (carotenes). Based on the differences in the reflectance spectra obtained for solid fat by-products, and melted and re-solidified fat (Figure 2), it is clear that the heat applied to melt the fat caused severe changes in its ultrastructure, which were not reversed when the fat re-solidified. 

## 5. Conclusions

The lipid profile of chicken fat by-products from the three farms was similar (with low coefficients of variation), despite factors associated with their different geographical locations (as long as the birds were fed a similar diet) which is very important finding for their potential application as a fat source in the formulation of meat products. The predominant fatty acids in chicken fat by-products were oleic, palmitic and linoleic acids, showing higher amount of unsaturated fatty acids than recorded for traditional fat sources used to make meat products. As regards the colour properties, chicken fat by-products had colour coordinate values that were in the range of those of chicken skin, which is the usual fat source in the meat industry. However, melting and re-solidification caused severe changes in the reflectance spectrum. In view of these results, chicken fat by-products could be used as fat ingredient in sausage formulations to partially or totally substitute traditionally used solid fat sources with their higher saturated fatty acid concentrations.

## Figures and Tables

**Figure 1 foods-09-01046-f001:**
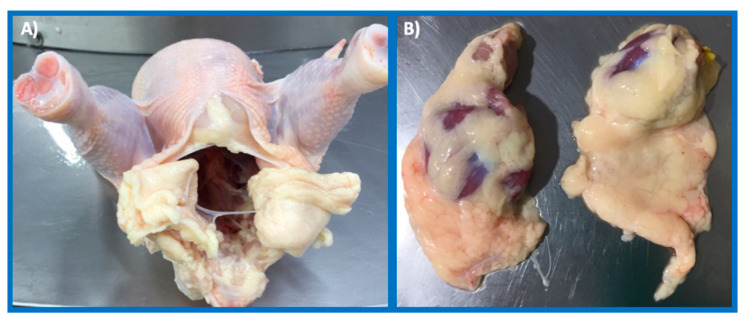
Chicken fat by-products: (**A**) Abdominal fat, (**B**) gizzard fat.

**Figure 2 foods-09-01046-f002:**
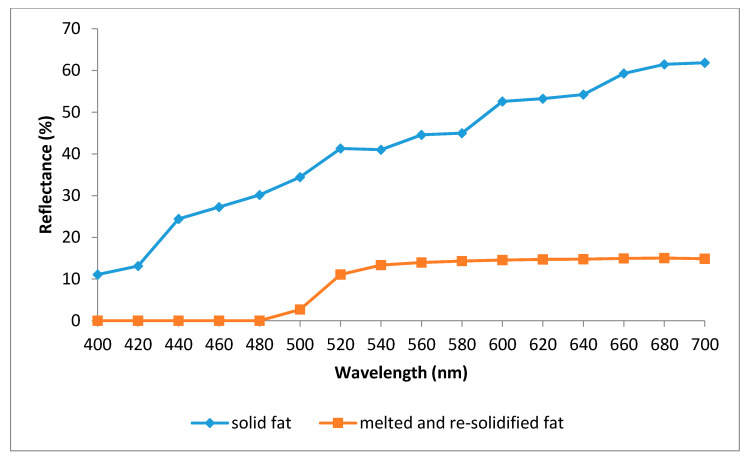
Reflectance spectra (400–700 nm) of the chicken fat by-products (solid fat and, melted and re-solidified fat).

**Table 1 foods-09-01046-t001:** Experimental finisher diets.

Ingredients (%)	Finisher Diets								
	1	2	3	4	5	6	7	8	9
Sorghum	10.00	10.00	10.00	10.00	10.00	10.00	10.00	10.00	10.00
Soy oil	2.95	2.92	3.24	3.20	4.00	4.00	4.00	4.00	3.50
Yellow corn	10.00	10.00	10.00	10.00	10.00	10.00	10.00	10.00	10.00
White corn	43.68	43.79	44.45	44.30	45.00	44.95	45.03	34.84	32.84
Corn gluten	4.16	4.26	4.38	4.85	4.72	4.09	4.11	-	-
Wheat	-	-	-	-	-	-	-	10.00	10.00
Bone flour	3.18	3.00	3.06	2.87	2.89	3.23	3.19	4.50	3.45
Soybean meal	7.61	7.61	11.17	11.01	14.94	14.68	14.77	11.77	11.51
Full fat soybean	12.50	12.50	10.00	10.00	4.49	5.16	5.03	10.00	14.99
Sunflower meal	2.00	2.00	-	-	-	-	-	-	-
Nutrients (% of diet)									
Protein	18.56	18.51	18.54	18.57	18.58	18.70	18.59	18.46	18.45
Lipids	8.46	8.97	8.41	8.97	8.30	8.24	8.16	8.28	8.25
Fibre	3.90	3.88	3.90	3.89	3.63	3.62	3.57	3.58	3.58
Minerals	3.32	3.37	3.26	3.26	3.26	3.20	3.22	3.34	3.33
Fatty acids (% total fat)									
C14:0 (Myristic acid)	0.19	0.19	0.18	0.17	0.15	0.16	0.16	0.22	0.20
C16:0 (Palmitic acid)	11.51	11.48	11.45	11.41	11.27	11.35	11.34	11.70	11.54
C16:1 (Palmitoleic acid)	0.86	0.84	0.91	0.89	1.04	1.05	1.05	1.06	0.90
C18:0 (Stearic acid)	4.04	4.00	4.01	3.97	3.97	4.04	4.03	4.32	4.15
C18:1 (Oleic acid)	24.90	24.86	24.93	24.88	24.97	25.03	25.03	24.85	24.36
C18:2 (Linoleic acid)	52.29	52.42	52.34	52.48	52.47	52.22	52.24	51.36	52.12
C18:3 (Linolenic acid)	5.62	5.62	5.57	5.57	5.43	5.46	5.45	5.81	6.09
C > 19	0.59	0.59	0.63	0.62	0.71	0.70	0.70	0.69	0.64

**Table 2 foods-09-01046-t002:** Productivity parameters of “Parrilleros” chickens in the three farms under study (*n* = 75 per farm).

Item ^1^/Farm	La Goleta	Villa Rita	La Nirvana
**FBW (kg)**	2.31 ± 0.02	2.37 ± 0.05	2.32 ± 0.04
**BWG (g/day)**	57.4 ± 1.23	57.3 ± 1.20	57.8 ± 1.68
**FI (kg)**	3.66 ± 0.07	3.80 ± 0.09	3.78 ± 0.08
**FCR (kg/kg)**	1.58 ± 0.03	1.59 ± 0.05	1.63 ± 0.04

*n* = number of birds (whole flock); ^1^ Each value represents the mean of 3 replicates (25 birds per pen). No significant differences (*p* < 0.0) were found between farms. FBW: final body weight; BWG: average of daily body weight gain; FI: feed intake; FCR: feed conversion ratio.

**Table 3 foods-09-01046-t003:** Lipid profile (% of total lipids) of chicken fat by-products from the three farms under study.

Fatty Acid	Common Name	Farms			Variation Coefficient (%)
		Villa Rita	La Goleta	La Nirvana	
C14:0	Myristic acid	0.52	0.50	0.50	3.9
C16:0	Palmitic acid	24.18	23.63	23.81	4.0
C16:1	Palmitoleic acid	5.01	4.83	5.16	3.6
C18:0	Stearic acid	5.69	6.00	5.92	5.4
C18:1*w*9	Oleic acid	36.15	36.83	35.31	6.0
C18:2*w*6	Linoleic acid	22.55	22.22	23.75	0.6
C18:3*w*3	Linolenic acid	1.46	1.49	1.64	1.0
∑ SFA		30.4	30.1	30.2	3.6
∑ MUFA		41.2	41.7	40.5	5.4
∑ PUFA		24.0	23.7	25.4	0.5
∑ PUFA/∑ SFA		0.8	0.8	0.8	

**Table 4 foods-09-01046-t004:** Colour parameters [(L*) lightness, (a*) redness, (b*) yellowness, (C*) chroma or saturation index, (h*) hue] of chicken fat by-products (solid fat and melted and re-solidified fat).

Chicken Fat by-Product	L*	a*	b*	C*	h*
Solid	71.52 ± 2.22a	3.44 ± 0.09 ^a^	24.65 ± 1.56 ^b^	24.89 ± 1.47 ^b^	82.06 ± 1.25 ^b^
Melted and re-solidified	40.26 ± 1.03b	0.96 ± 0.03 ^b^	65.89 ± 1.02 ^a^	65.9 ± 2.14 ^a^	89.17 ± 1.22 ^a^

^a,b^: different letters indicate significant differences (*p* < 0.05). *n* = 39.

**Table 5 foods-09-01046-t005:** Lipid profile (%) of traditional fat sources in the meat industry, according to the literature, and of chicken fat by-products analyzed in this work.

Fatty Acid	Common Name	Beef Tallow ^(1)^	Pork Lard ^(2)^	Poultry Skin ^(3)^	Chicken Fat by-Products
C14:0	Myristic acid	1–1.5	1–1.5	–	0.51
C16:0	Palmitic acid	24–28	24–28	20–24	23.87
C16:1	Palmitoleic acid	2–3	2–3	5–9	5.00
C18:0	Stearic acid	20–24	13–14	4–6	5.87
C18:1*w*9	Oleic acid	40–43	43–47	33–44	36.10
C18:2*w*6	Linoleic acid	2–4	8–11	18–20	22.84
C18:3*w*3	Linolenic acid	<1	<1	1–2	1.53
∑ SFA		46–55	38–43.5	25–31.5	30.23
∑ MUFA		42–46	45–50	38–53	41.13
∑ PUFA		2–4	8–11	19–22	24.37
∑ PUFA/∑ SFA		<0.1	0.3	0.8	0.8

^(1)^ Mottram et al. [24]; Alm [25]. ^(2)^ Mottram et al. [24]; Ospina-E et al. [22]; Alm [25]. ^(3)^ Sheu & Chen [20]; Feddern et al. [17]; Alm [25].

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
