# Peer review of "Quality of Chicken Fat by-Products: Lipid Profile and Colour Properties"

_foods, 2020, doi:10.3390/foods9081046_

Round 1

Reviewer 1 Report

The paper is interesting and provides useful information on chicken fat by-products. Even if the analysis of the chickens’ productive parameters is not the main goal of the present paper, I suggest some improvements to better present these data.

Line 69-73: I guess these are commercial farm, please provide an average number of birds reared in each farm, also average density and photoperiod.

Table 1: in materials and method you stated that birds where reared in 3 farms but you show 9 finishing diets, can you explain this? I supposed that each farm used only one finishing diet.

Line 80-83: please give more details about slaughtering: did birds stunned (i.e. electrical stunning) before slaughtering?

Do you state that 13 birds/farm were slaughtered, so in total 39 birds? Please report the total number of birds used for the study.

Table 2: I see you reported standard deviation for FI and FCR: how did you calculate these parameters? Are birds reared in separate pens with several replicates? This is the only way to have a standard deviation for these parameters. Lease clarify this in materials and methods. Please provide n=

Table 3: It’s a pity you don’t have replicates for fatty acid composition for each farm in order to perform statistical analysis

Table 4: in this table, you show standard deviation, which was the "n"?

Line 160: “as IT can be seen”: IT is missing

Table 5: would be useful to include also your average results (average on the 3 farms)

Author Response

We thank all your comments and suggestions that allow us to clarify the message of our paper.

The paper has been carefully revised and language and grammatical errors have been corrected.

I am going to answer all your comments point by point. Your comments are in blue and our answers in black color.

The paper is interesting and provides useful information on chicken fat by-products. Even if the analysis of the chickens’ productive parameters is not the main goal of the present paper, I suggest some improvements to better present these data.

Yes, it is true that the analysis of the chickens’ productive parameters is not the main goal of our paper but of course, we are going to take into account all your suggestions to improve the presentation and diffusion of our data

Line 69-73: I guess these are commercial farm, please provide an average number of birds reared in each farm, also average density and photoperiod.

This information has been included L73-75

“The average number of birds reared and the average density (bird/m2) in each farm were: Villa Rita, 78,432 and 12.9, respectively; La Nirvana, 314,200 and 13.0, respectively; and La Goleta 1,132,002 and 12.4, respectively. In all the farms the photoperiod was 12 h (± 30 min) (12L/12D)”.

Table 1: in materials and method you stated that birds where reared in 3 farms but you show 9 finishing diets, can you explain this? I supposed that each farm used only one finishing diet.

Not exactly. It has been explained in Lines 76-78 “Nine different “finisher” diets were assessed (Table 1) depending on the availability of raw materials and prices in attempt to minimize costs for the companies, while maintaining the same nutritional levels

As you have said before, this work was made in commercial farms and the company have formulated these 9 diets with the same nutritional levels and so the farms can use one or other depending on the availability of raw materials and prices in attempt to minimize costs for the company.

Line 80-83: please give more details about slaughtering: did birds stunned (i.e. electrical stunning) before slaughtering?

Yes of course, they were stunned by electrical stunning before slaughtering. It has been included Line 88

Do you state that 13 birds/farm were slaughtered, so in total 39 birds? Please report the total number of birds used for the study.

Yes, 13 birds/farm were slaughtered, so in total 39 birds. It has been clarified in the paper

Table 2: I see you reported standard deviation for FI and FCR: how did you calculate these parameters? Are birds reared in separate pens with several replicates? This is the only way to have a standard deviation for these parameters. Lease clarify this in materials and methods. Please provide n=

It is true that the calculation of the productive parameters is not clear enough in the text. Now it has been completed.

“In each farm, 75 birds were selected for the experiment (kept in pens on litter) and were divided into 3 replications with 25 birds per group. Each bird had a padlock badge for identification during measurements”.

This information has been also included in table 2

Table 3: It’s a pity you don’t have replicates for fatty acid composition for each farm in order to perform statistical analysis

Yes, I understand but you know that in some cases we have to fight with the industry to obtain the samples in good conditions and also the price of these analysis in Colombia is very high.

Table 4: in this table, you show standard deviation, which was the "n"?

In this case the n=39 because no differences were stablished between farms

Line 160: “as IT can be seen”: IT is missing

Sorry for the error. “It” has been included

Table 5: would be useful to include also your average results (average on the 3 farms)

Yes, it is a good idea. Now these values have been included in Table 5

Reviewer 2 Report

Manuscript title: Quality of chicken fat by-products: Lipid profile and colour properties

The aim of the manuscript was to determinate the fatty acid profile and colour properties of poultry fat by-products (in this case, abdominal and gizzard fat) and to assess whether these properties remain stable and whether they depend on the farm conditions (feeding and geographical location).

The applied methods ale very poor. In these case also reological properties of the fats should be measured. Lipid oxidation of the fats (TBARS) would also enriched the analysis. In the research analysis of fatty acid profiles and colour parameters are in my opinion insufficient.

Detailed comments

Line 18: should be „colour” instead of „color”

Line 26: keyword „healthy fat” should be delated

Table 1 – the names of fatty acids should be added

There is lack of statistical analysis in the table

The numer of repetition should be added in the fatty acid profile analysis and in the colour analysis also

In the table 2 the letteres „a” should be delated. There are any statistically differences so it is not needed.

Line 134 Why’’8” different diets? In the table 1 there is 9 diets.

Line 141, 154  It should be by-products instead of byproducts

Line 152 It should be had statistically higher

Author Response

Responses to Reviewer 2

We thank all your comments and suggestions that allow us to clarify the message of our paper.

The paper has been carefully revised and language and grammatical errors have been corrected.

I am going to answer all your comments point by point. Your comments are in blue and our answers in black color.

Manuscript title: Quality of chicken fat by-products: Lipid profile and colour properties.

The aim of the manuscript was to determinate the fatty acid profile and colour properties of poultry fat by-products (in this case, abdominal and gizzard fat) and to assess whether these properties remain stable and whether they depend on the farm conditions (feeding and geographical location).

Yes, it was our objective in this work

The applied methods ale very poor. In these case also reological properties of the fats should be measured. Lipid oxidation of the fats (TBARS) would also enriched the analysis. In the research analysis of fatty acid profiles and colour parameters are in my opinion insufficient.

I understand this comment but it was made with the final objective that these chicken fat by-products were used as partial fat replacement in chicken sausages. These sausages were successfully produced with a 40% fat replacement without significant differences in texture, color, lipid oxidation, sensorial analysis and shelf-life. This work has been published and cited in this paper

Peña-Saldarriaga, L.; Pérez-Alvarez, J.A.; Fernández-López, J. Quality properties of chicken emulsion-type sausages formulated with chicken fatty byproducts. Foods 2020, 9, 507.

Detailed comments

Line 18: should be „colour” instead of „color”

  1. It has been corrected

Line 26: keyword „healthy fat” should be delated

  1. It has been deleted

Table 1 – the names of fatty acids should be added

Ok. The names of fatty acids have been added as you suggested

The numer of repetition should be added in the fatty acid profile analysis and in the colour analysis also

  1. These analysis were made in triplicate, now It has been included

In the table 2 the letteres „a” should be delated. There are any statistically differences so it is not needed.

It is true, now they have been deleted

Line 134 Why’’8” different diets? In the table 1 there is 9 diets.

I’m sorry for the error in the text. The number of diets was 9 as has been shown in the table 1 and now it has been corrected

Line 141, 154  It should be by-products instead of byproducts

Both errors in the spelling have been corrected

Line 152 It should be had statistically higher

I don’t understand this comment.

Round 2

Reviewer 2 Report

I accept the Authors' corrections. 

Line 152 It should be had statistically higher
I don’t undeLine 152 It should be had statistically higher
I don’t understand this comment

In this line it should be added "statistically"

Author Response

Responses to Reviewer 2

Thanks for accepting our corrections.

We appreciate your clarifications regarding one of your previous corrections. Sorry we did not understand the correction from the beginning.

Line 152 It should be had statistically higher

In this line it should be added "statistically"

Now it is clear and “statistically” has been added in Line 175 (corresponding to the initial line 152)
